# Relationships between socioeconomic position and objectively measured sedentary behaviour in older adults in three prospective cohorts

Richard John Shaw,[1] Iva Čukić,[2] Ian J Deary,[2] Catharine R Gale,[2,3] Sebastien FM Chastin,[4,5] Philippa M Dall,[4] Dawn A Skelton,[4] Geoff Der,[1,2] on behalf of the Seniors USP Team

[1]MRC/CSO Social and Public Health Sciences Unit, University of Glasgow, Glasgow, UK
[2]Department of Psychology Centre for Cognitive Ageing and Cognitive Epidemiology, University of Edinburgh, Edinburgh, UK
[3]MRC Lifecourse Epidemiology Unit, University of Southampton, Southampton, UK
[4]Institute for Applied Health Research, School of Health and Life Sciences, Glasgow Caledonian University, Glasgow, UK
[5]Department of Movement and Sports Sciences, Faculty of Medicine and Health Science, Ghent University, Ghent, Belgium

**Correspondence to**
Dr. Richard John Shaw;
dr.richard.shaw@gmail.com

## ABSTRACT

**Objectives** To investigate whether sedentary behaviour in older adults is associated with a systematic and comprehensive range of socioeconomic position (SEP) measures across the life course. SEP measures included prospective measures of social class, income, educational qualifications and parental social class and contemporaneous measures of area deprivation.

**Setting** Glasgow and the surrounding (West of Scotland) combined with Edinburgh and the surrounding area (the Lothians).

**Participants** Community-dwelling adults aged around 79, 83, and 64 years from, respectively, the Lothian Birth Cohort 1936 (LBC1936) (n=271) and the 1930s (n=119) and 1950s (n=310) cohorts of the West of Scotland Twenty-07 study.

**Primary outcome measure** Sedentary behaviour was measured objectively using an activPAL activity monitor worn continuously for 7 days and used to calculate percentage of waking time spent sedentary.

**Results** Among retired participants, for most cohort and SEP combinations, greater social disadvantage was associated with increased sedentary time. For example, in the Twenty-07 1930s cohort, those most deprived on the Carstairs measure spent 6.5% (95% CI 0.3 to 12.7) more of their waking time sedentary than the least deprived. However, for employed people, the relationship between SEP and sedentary behaviour was much weaker. For example, in terms of social class differences, among the retired, the most disadvantaged spent 5.7% more waking time sedentary (95% CI 2.6% to 87%), whereas among the employed, there was effectively no difference (−0.5%; 95% CI −9.0 to 8.0).

**Conclusions** Diverse SEP measures were associated with increased sedentary behaviour among retired people. There was little evidence for a relationship between SEP measures and sedentary behaviour among employed older adults. Prior to retirement, the constraints of the workplace may be masking effects that are only apparent at weekends.

► Sedentary behaviour is objectively measured using an activPAL monitor that correctly identifies posture.
► Data are available for an entire 7-day period, thus minimising any systematic variation over the course of the week.
► Our study is the first that we are aware of that uses a comprehensive and systematic range of socioeconomic position measures across the life course.
► Due to attrition, the sample is more socioeconomically advantaged than the original populations from which it was drawn.

posture,[1] is emerging as a modifiable risk factor for poor health independent of physical activity.[2 3] There is evidence for associations with: mortality,[3–5] cancer incidence,[3] diabetes,[6] bone density[7] and falls.[8] Sedentary behaviour increases with age.[9] On average, sedentary time represents 65%–80% of an older adult's waking day[10] and 67% of older adults spend in excess of 8.5 hours per day sitting.[11] Reducing sedentary behaviour may, therefore, be an important health message for older adults.

Socioeconomic position (SEP) has been described as a fundamental cause of poor health and health inequalities.[12] SEP represents flexible resources that shape people's opportunities and provide support for their efforts to engage health-enhancing behaviours.[12] As such, it is a multidimensional concept[13] with different aspects of socioeconomic position being salient for different health outcomes and the importance of those aspects varying across the life course.[14]

The current literature on the socioeconomic determinants of sedentary behaviour in older adults is very limited.[9 15] Of the few

Sedentary behaviour, defined as energy expenditure ≤1.5 metabolic equivalents while awake and in a sitting or reclining

studies that exist, most rely on self-reported measures of TV viewing and screen time.[16–22] However, these represent only two of the circumstances in which people might be sedentary. Moreover, self-reported measures of sedentary behaviour have only low to moderate validity[23] and, therefore, attenuate relationships. They are also subject to recall and social desirability bias.

To our knowledge, only five studies have examined the association of SEP with objectively measured sedentary behaviour in older people, and these have all used Acti-Graph accelerometers worn on the waist,[24–28] which do not accurately record posture.[29] Moreover, these studies did not explicitly focus on SEP as a potential determinant of sedentary behaviour. In short, while results from younger adults suggest SEP could be an important predictor of sedentary behaviour,[30] the situation for older adults remains unclear.

In this study, we investigate whether SEP is an important determinant of sedentary behaviour among older people. We use data from three Scottish cohorts aged in their 60s, 70s and 80s. Sedentary behaviour is measured using the activPAL monitor worn continuously for 7 days. As the participants are drawn from existing, long-standing cohorts, our study includes a diverse range of prospective indicators that capture many aspects of socioeconomic position including parental social class, education, household social class, neighbourhood deprivation, housing tenure, income and subjective social status.

## METHODS

### Participants

This study, Seniors USP (Understanding Sedentary Patterns), comprises subsamples of the Lothian Birth Cohort 1936 (LBC1936) and the West of Scotland Twenty-07 study (Twenty-07). Full details for these studies are available elsewhere.[31 32] The Twenty-07 study itself comprises three age cohorts, although only the two oldest are included here (hereafter, referred to as the 1930s and 1950s cohorts according to their decade of birth). Data for the main Twenty-07 study were collected in five waves of interviews between 1987 and 2008. LBC1936 is an ongoing cohort study of older people that began in 2004 as a follow-up to the Scottish Mental Survey 1947.

Data specific to this study, including objective sedentary behaviour, were collected between November 2014 and April 2016 when the mean ages of the cohorts were around 64, 79 and 83 years.

To be eligible to take part in Seniors USP, participants had to live in the community and needed sufficient cognitive ability to be able to provide informed consent and also a sufficiently good memory to complete sleep diaries. Beyond these minimum requirements, people were not excluded due to physical or mental impairments. Twenty-07 participants were eligible if they lived within the greater Glasgow area. All eligible people in the 1930s cohort were approached, and a random sample of eligible people in the 1950s cohort was selected. Consecutive

recruits to wave 4 of LBC1936 were invited to join Seniors USP until the target sample size of 300 was achieved.

Participants of the original Twenty-07 study were representative of the communities from which they were drawn.[33] The LBC1936 participants were drawn from Scottish Mental Survey, which was a whole population sample but are advantaged with respect to SEP, though all groups are represented. Due to attrition, participants in Seniors USP show some differences from the cohorts from which they are drawn. For the 1950s cohort, the analysed sample was more advantaged with respect to the lifetime and parental social class, subjective social position, educational qualifications, minimum school leaving age, tenure and car ownership. For the 1930s cohort, the analysed sample had an older school leaving age and higher income, but there were no significant differences for the other SEP measures. For LBC1936, the analysed sample were more advantaged with respect to tenure, educational qualifications and minimum school leaving age, but there were no significant differences in lifetime and parental social class.

### Sedentary behaviour

Sedentary behaviour was measured using the activPAL monitor (activPAL3c, PAL Technologies Ltd, Glasgow, UK), a small and light (53x35×7 mm; 15 g) tri-axial accelerometer, worn attached to the anterior thigh of the dominant leg with a waterproof dressing. The monitor samples acceleration at 20 Hz, which is then categorised into time spent in sedentary or upright posture based on thigh inclination. Additionally time spent walking is identified based on thigh acceleration. The monitor provides accurate and reliable measurement of sedentary behaviour.[34 35] Participants were initially interviewed for basic sociodemographic and health information and were then asked to wear the activPAL continuously for 7 days, while going about their usual daily activities, including overnight and during bathing or swimming. Participants also kept a diary reporting the time they fell asleep the previous night and the time they woke up for each day of monitoring. Self-reported wake and sleep times from the diary were used to isolate waking data for analysis. Participants without a full 7 days were excluded in order to avoid making any assumptions about wear time. There was little evidence to suggest that included and excluded participants differed on any of the SEP measures.

The outcome measure is the percentage of waking time spent sedentary, averaged over the 7 days (hereafter, sedentary time).

### Socioeconomic position

We include prospective measures of SEP based on the three major axes of social stratification: education, occupation and income. In addition, there are two measures of area deprivation, and one measure each of tenure, car ownership and subjective social position. Full details of these measures including when they were collected and how they differed between LBC1936 and the Twenty-07

are given in the online supplementary appendix 1. A brief description follows.

Occupation-based measures are parental social class (professional/intermediate/skilled/semiskilled/ unskilled) and lifetime social class (professional/managerial/skilled non-manual/skilled manual/semi-skilled or unskilled).

Education measures are: highest educational qualification (none/basic/degree or professional) and whether or not left school at minimum leaving age.

The income measure is net household income equivalised to adjust for household composition using the McClements Scales.[36]

Area deprivation measures include the Scottish Index of Multiple Deprivation 2012 (SIMD)[37] and Carstairs deprivation score.[38] Both measures are based on the datazone of the participant's residence. The SIMD comprises seven domains: income, employment, education, health, crime, housing and access to facilities, that are combined to create an overall deprivation score. The Carstairs deprivation score is based on four measures from the 2011 census: car ownership, male unemployment, overcrowding and low social class.[39]

Our measure of housing tenure contrasts home owners with others and likewise our measure of car ownership.

Subjective social status was assessed using a self-anchoring scale in the form of a 10-rung ladder representing society,[40] and participants were asked to indicate where they considered themselves to be in relation to others in Britain.

## Statistical methods

Differences between the cohorts were identified using $\chi^2$ tests for the ordinal and binary measures and analysis of variance for the continuous measures. Associations between SEP and outcome measures were estimated using linear regression.

In order to facilitate comparisons between SEP measures made on different scales of measurement, we used the slope index of inequality (SII).[41] This involves rescaling the SEP measures to fractional ranks, that is, ranking them and dividing by the sample size. Where there are ties in the data, the mid-rank is assigned. For the area deprivation measures, ranks are available for the whole of Scotland. For other measures, ranking is cohort specific. For highest educational qualifications, all five ordinal categories (see online supplementary appendix 1) that were available for each cohort were used to derive the SII, while three categories were used for presentation in tables. For all measures, higher ranks are assigned to greater disadvantage, and the SII can therefore be interpreted as the difference in outcome between the hypothetically most and least disadvantaged.

Additional analyses based on the original scores are presented in the online supplementary appendix 1.

All analyses were conducted using Stata V.13.1.

## RESULTS

Seven hundred and seventy-three participants took part: 340, 129 and 304 each from the 1950s cohort, 1930s and LBC1936, respectively. Of these, 700 (91%) provided seven full days of activPAL and sleep diary data. Two participants were excluded because they did not return activPAL devices. Seven were excluded because there was incomplete sleep diary data. Five were excluded due to poor activPAL quality, assessed using a graphical inspection of the data and 59 due to having insufficient days of data. The reasons for insufficient days of data were varied and not always reported. Eight removed devices due to skin irritation, in other cases the device had fallen off, become wet or had been removed for a variety of other reasons. We analysed only those who had full data so no assumptions about wear time would have to be made.

Previous research[42] has suggested that the social patterning of physical activity differs before and after retirement, and preliminary analysis of sedentary behaviour in the 1950s cohort showed a similar pattern. Consequently, we have divided this cohort into those still employed, including the semi-retired, versus those no longer employed. We refer to the latter as 'retired' even though not all would consider themselves formally retired.

Table 1 shows the breakdown of the sample by SEP and demographic measures. There were significant cohort differences in all the SEP measures (p<0.01), except parental social class, reflecting differences between the areas from which they are drawn and secular changes in the occupational structure. For all SEP measures the 1930s cohort are the most disadvantaged.

Within the 1950s cohort, there were few differences in SEP measures between the employed and retired. The only significant differences were for gender (0.001), only 36.4% of 1950s employed were female compared with 62.5% of the retired, and housing tenure for which 14.0% of the retired were renting compared with only 4.6% of the employed.

Table 2 shows the mean and SD of sedentary time by cohort. As might be expected given their ages, the 1930s cohort are the most sedentary, on average spending 68.2% of the day sedentary. Among the 1950s cohort, the retired have similar levels to LBC1936 (62.2% and 62.5%, respectively), whereas the employed have the lowest level (58.3%). Despite LBC1936 being closer in age to the 1930s cohort, their sedentary time is more like the 1950s retired. As the 1930s cohort is the most deprived and LBC1936 the least, it may be that the overall cohort differences reflect the differences in SEP as well as age differences.

Throughout the table, there are numerous examples of SEP differences in sedentary time, although the small numbers in some categories, especially the extremes of some social class measures, need to be borne in mind.

A comparison of the analysed sample with the remainder of those invited showed a number of differences (data not shown). For LBC1936, there were no

**Table 1** Descriptive statistics by cohort and employment status during Seniors USP

| | Twenty-07 1950s employed | | Twenty-07 1950s retired | | Twenty-07 1930s All | | LBC1936 All | |
|---|---|---|---|---|---|---|---|---|
| | n=110 | | n=200 | | n=119 | | n=271 | |
| | n | % | n | % | n | % | n | % |
| **Tenure** | | | | | | | | |
| Own or mortgage | 104 | 94.6 | 171 | 85.5 | 95 | 79.8 | 259 | 95.6 |
| Renting or other | 5 | 4.6 | 28 | 14.0 | 21 | 17.7 | 12 | 4.4 |
| Missing | 1 | 0.9 | 1 | 0.5 | 3 | 2.5 | 0 | 0.0 |
| **Car ownership** | | | | | | | | |
| No | 9 | 8.2 | 27 | 13.5 | 42 | 35.3 | | |
| Yes | 100 | 90.9 | 172 | 86.0 | 74 | 62.2 | | |
| Missing | 1 | 0.9 | 1 | 0.5 | 3 | 2.5 | | |
| **Lifetime social class** | | | | | | | | |
| I Professional | 27 | 24.6 | 45 | 22.5 | 18 | 15.1 | 69 | 25.5 |
| II Managerial | 60 | 54.6 | 94 | 47.0 | 47 | 39.5 | 101 | 37.3 |
| III Skilled non-manual | 19 | 17.3 | 43 | 21.5 | 31 | 26.1 | 50 | 18.5 |
| III Skilled manual | 3 | 2.7 | 11 | 5.5 | 16 | 13.5 | 38 | 14.0 |
| IV/V Semiskilled/ unskilled | 1 | 0.9 | 7 | 3.5 | 7 | 5.5 | 10 | 3.7 |
| Missing | 0 | 0.0 | 0 | 0.0 | 0 | 0.0 | 3 | 1.1 |
| **Highest qualification** | | | | | | | | |
| None | 10 | 9.1 | 15 | 7.5 | 34 | 28.6 | 36 | 13.3 |
| Basic | 65 | 59.1 | 96 | 48.0 | 61 | 51.3 | 133 | 49.1 |
| Degree or professional qualification | 35 | 31.8 | 89 | 44.5 | 24 | 20.2 | 102 | 37.6 |
| **Minimum school leaving age** | | | | | | | | |
| Stayed past min age | 69 | 62.7 | 131 | 65.5 | 61 | 51.3 | 144 | 53.1 |
| Left minimum or earlier | 40 | 36.4 | 68 | 34.0 | 55 | 46.2 | 127 | 46.9 |
| Missing | 1 | 0.9 | 1 | 0.5 | 3 | 2.5 | 0 | 0 |
| **Parental social class** | | | | | | | | |
| I Professional | 6 | 5.5 | 16 | 8.0 | 6 | 5.0 | 20 | 7.4 |
| II Intermediate | 21 | 19.1 | 33 | 16.5 | 14 | 11.8 | 54 | 19.9 |
| III Skilled (manual and non-manual) | 54 | 49.1 | 98 | 49.0 | 61 | 51.3 | 136 | 50.2 |
| IV Partly skilled | 15 | 13.6 | 32 | 16.0 | 16 | 13.5 | 28 | 10.3 |
| V Unskilled | 12 | 10.9 | 16 | 8.0 | 12 | 10.1 | 15 | 5.5 |
| Missing | 2 | 1.8 | 5 | 2.5 | 10 | 8.4 | 18 | 6.6 |
| **Gender** | | | | | | | | |
| Male | 70 | 63.6 | 75 | 37.5 | 54 | 45.4 | 140 | 51.7 |
| Female | 40 | 36.4 | 125 | 62.5 | 65 | 54.6 | 131 | 48.3 |
| **SIMD** | | | | | | | | |
| Mean | 0.37 | | 0.42 | | 0.45 | | 0.22 | |
| SD | 0.30 | | 0.32 | | 0.33 | | 0.25 | |
| **Carstairs** | | | | | | | | |
| Mean | 0.39 | | 0.44 | | 0.47 | | 0.3 | |
| SD | 0.29 | | 0.31 | | 0.32 | | 0.25 | |

**Table 1**  Continued

| | Twenty-07 1950s employed | | Twenty-07 1950s retired | | Twenty-07 1930s All | | LBC1936 All | |
|---|---|---|---|---|---|---|---|---|
| | n=110 | | n=200 | | n=119 | | n=271 | |
| | n | % | n | % | n | % | n | % |
| **Household net income** | | | | | | | | |
| Mean | 5.78 | | 6.22 | | 4.01 | | | |
| SD | 3.09 | | 3.92 | | 2.39 | | | |
| Missing | 11 | | 15 | | 22 | | | |
| **Subjective social position** | | | | | | | | |
| Mean | 6.52 | | 6.14 | | 6.1 | | | |
| SD | 1.47 | | 1.70 | | 1.62 | | | |
| Missing | 2 | | 3 | | 7 | | | |
| **Age** | | | | | | | | |
| Mean | 64.4 | | 64.7 | | 83.4 | | 79.0 | |
| SD | 0.88 | | 0.89 | | 0.62 | | 0.44 | |

SIMD, Scottish Index of Multiple Deprivation; Seniors USP, Understanding Sedentary Patterns.

significant differences in lifetime and parental social class, gender or self-rated health, but the analysed sample was more advantaged with respect to tenure, educational qualifications and minimum school leaving age. For the 1950s cohort, the analysed sample were more advantaged with respect to all the prospective SEP measures and self-rated health, but there was no difference in gender. For the 1930s cohort, the analysed sample had better self-rated health and older school leaving age, but there were no significant differences for the other SEP measures.

### Analysis of sedentary time

Figure 1 presents the results of linear regression analyses regressing sedentary time on the SEP measures. Separate results are shown for: the LBC1936 cohort, the 1930s cohort, the employed and retired subgroups of the 1950s cohort and the 1950s retired group combined with the two older cohorts. In each case, regression coefficients and their 95% CIs are shown. For the binary SEP measures, the effect is simply the difference in sedentary time between the two groups. For the other SEP measures, the effect is the SII and is interpretable as the difference in sedentary time between the most and least deprived. The overall pattern is one of more disadvantaged SEP being associated with greater sedentary time. The exceptions are the employed group and parental social class where there are no clear or consistent patterns. In terms of statistical significance, when the retired people in the 1950s cohort was combined with the two older cohorts all SEP measures were significantly and positively related to sedentary time, the largest SIIs being found for subjective social position (7.6%, 95% CI 3.5 to 11.7) and Carstairs deprivation (6.6%, 95% CI 3.6 to 9.5). When these cohorts were investigated separately, relationships between sedentary time and SEP were found for most cohort and SEP combinations. In some cases, particularly for the 1930s

cohort, the CIs crossed zero, although effects were in the same direction and of similar magnitude. There is limited evidence that the effects of SEP varied by cohort. There was only one significant interaction between an SEP measure and cohort for the retired cohorts, which was for parental social class (p=0.0043).

In contrast to the retired cohorts, none of the SEP measures were significantly associated with sedentary time for the employed in the 1950s cohort. Furthermore, half of the coefficients were below zero suggesting that there was no evidence of a relationship between sedentary time and SEP measures in general.

### Sensitivity tests and additional analyses

Additional analyses are presented in the appendices. Table A1 in online supplementary appendix 1 includes the regression coefficients for the SEP measures in their original form as shown in tables 1 and 2. Broadly these are consistent with the results produce by the SII; however, far fewer measures are significant, partly reflecting the reduced statistical power of categorical variables. It should also be noted that there was a non-linear relationship between income and sedentary time among the retired subgroup of the 1950s cohort. This indicates that sedentary time falls with increases in net equivalised household incomes up to about £1000 a month beyond which further income makes little difference.

In online supplementary appendix 2, appendix table A2 shows p values for tests of differences between weekdays and weekends in the association between SEP measures and sedentary time. The results for weekdays are presented in online supplementary appendix figure A1 and weekends in online supplementary figure A2. We find little evidence of substantive or systematic differences between weekdays and weekends for retired people. For employed people in the 1950s cohort, the

**Table 2** Mean and SD for and percent waking time spent sedentary

| | Twenty-07 1950s Employed | | Twenty-07 1950s Retired | | Twenty-07 1930s All | | LBC1936 All | |
|---|---|---|---|---|---|---|---|---|
| | Mean | SD | Mean | SD | Mean | SD | Mean | SD |
| **All members** | | | | | | | | |
| Overall | 58.3 | 11.2 | 62.2 | 10.3 | 68.2 | 10.9 | 62.5 | 10.4 |
| **Tenure** | | | | | | | | |
| Own or mortgage | 58.1 | 11.4 | 61.5 | 10.1 | 67.8 | 10.8 | 62.1 | 10.2 |
| Renting or other | 63.5 | 6.6 | 66.1 | 10.9 | 70.8 | 11.3 | 70.7 | 11.9 |
| **Car ownership** | | | | | | | | |
| No | 58.1 | 11.5 | 66.0 | 9.7 | 71.2 | 10.7 | | |
| Yes | 58.4 | 11.2 | 61.6 | 10.3 | 66.7 | 10.8 | | |
| **Lifetime social class** | | | | | | | | |
| I Professional | 57.4 | 13.3 | 60.5 | 9.7 | 62.5 | 11.2 | 60.3 | 10.1 |
| II Managerial | 59.5 | 10.2 | 61.6 | 10.4 | 68.8 | 11.5 | 62.9 | 10.0 |
| III Skilled non-manual | 56.1 | 11.2 | 64.1 | 9.7 | 68.3 | 9.4 | 60.7 | 10.5 |
| III Skilled manual | 60.1 | 14.7 | 66.2 | 12.6 | 69.4 | 10.0 | 65.7 | 10.8 |
| IV/V Semiskilled unskilled | 58.3 | – | 66.7 | 13.0 | 76.0 | 10.7 | 66.9 | 8.9 |
| **Highest achieved qualification education** | | | | | | | | |
| None | 54.7 | 13.9 | 62.6 | 11.4 | 71.6 | 10.0 | 64.8 | 11.7 |
| Basic | 58.5 | 11.0 | 64.1 | 9.9 | 66.1 | 11.4 | 62.5 | 10.2 |
| Degree/prof qualification | 59.1 | 10.8 | 60.1 | 10.4 | 68.9 | 10.2 | 61.7 | 10.1 |
| **Minimum school leaving age** | | | | | | | | |
| Stayed pass min age | 57.4 | 11.4 | 60.3 | 9.8 | 67.3 | 10.2 | 61.3 | 10.1 |
| Left minimum or earlier | 60.0 | 10.8 | 65.8 | 10.5 | 69.4 | 11.6 | 63.8 | 10.5 |
| **Parental social class** | | | | | | | | |
| I Professional | 60.7 | 11.8 | 58.4 | 9.9 | 69.7 | 7.7 | 66.5 | 7.2 |
| II Intermediate | 54.6 | 13.1 | 56.1 | 9.8 | 62.7 | 12.3 | 62.2 | 9.4 |
| III Skilled (manual and non-manual) | 59.6 | 10.1 | 63.5 | 10.2 | 69.4 | 5.0 | 62.5 | 9.6 |
| IV Partly skilled | 55.7 | 11.5 | 65.6 | 10.0 | 69.0 | 14.1 | 63.2 | 10.4 |
| V Unskilled | 62.3 | 11.0 | 63.3 | 8.8 | 68.6 | 15.8 | 60.3 | 11.8 |
| **Gender** | | | | | | | | |
| Male | 60.4 | 10.9 | 62.7 | 9.1 | 68.0 | 12.0 | 64.8 | 9.8 |
| Female | 54.8 | 10.8 | 61.9 | 11.0 | 68.5 | 10.0 | 60.1 | 10.5 |
| **SIMD overall** | | | | | | | | |
| <50 Percentile | 57.4 | 11.4 | 60.4 | 9.6 | 67.0 | 10.6 | 61.9 | 10.2 |
| >50 Percentile | 59.4 | 10.9 | 63.9 | 10.7 | 69.4 | 11.2 | 63.1 | 10.5 |
| **Carstairs score** | | | | | | | | |
| <50 Percentile | 57.7 | 11.8 | 60.4 | 9.5 | 66.8 | 11.1 | 60.9 | 9.6 |
| >50 Percentile | 59.1 | 10.4 | 63.8 | 10.8 | 69.6 | 11.6 | 64.2 | 10.9 |
| **Income** | | | | | | | | |
| >50 Percentile | 57.6 | 11.5 | 60.7 | 10.0 | 67.2 | 11.8 | | |
| <50 Percentile | 59.0 | 10.6 | 64.0 | 10.6 | 71.1 | 10.4 | | |
| **Subjective social position** | | | | | | | | |
| >50 Percentile | 59.5 | 11.0 | 60.2 | 9.3 | 66.4 | 10.6 | | |
| <50 Percentile | 57.2 | 11.5 | 64.5 | 11.0 | 69.3 | 11.4 | | |

LBC1936, Lothian Birth Cohort 1936; SIMD, Scottish Index of Multiple Deprivation.

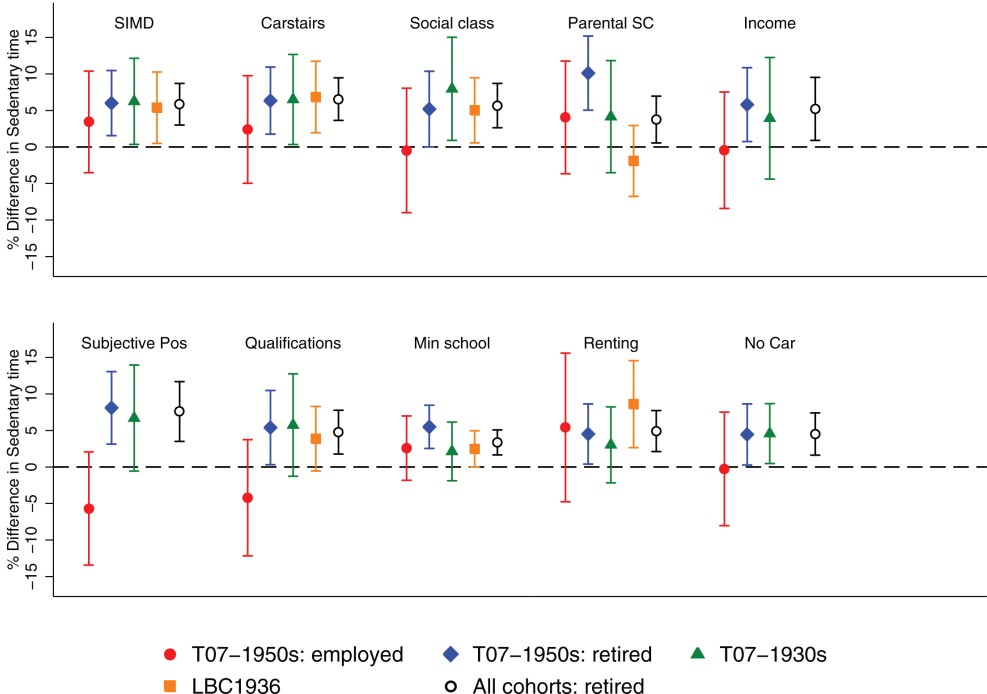

**Figure 1** Inequalities in sedentary time by 10 SEP measures for three Scottish cohorts. The inequalities represent SII (with 95% CIs) for each of the following SEP measures (unless otherwise stated in parentheses) the Scottish Index of Multiple Deprivation, Carstairs deprivation score, lifetime social class, parental social class, equivalised household income, subjective social position, highest educational qualification, school leaving age (stayed on beyond minimum reference), housing tenure (binary measure—owners reference) and car ownership (binary measure—car owners reference). A positive difference (point above the dotted line) indicates that a socioeconomically disadvantaged person is more sedentary a negative difference (point below the line) indicates the reverse. LBC1936, Lothian Birth Cohort 1936; SEP, socioeconomic position; SIMD, Scottish Index of Multiple Deprivation; T07, Twenty-07 study.

majority of SEP measures showed a different association with sedentary behaviour on weekdays from that on weekends. The pattern was one whereby the employed were more similar to the retired at weekends than during weekdays. Some measures, highest qualification and subjective social position may be associated with reduced sedentary behaviour during the week and show no association at weekends, while SIMD deprivation and parental social class were associated with increased sedentary behaviour at weekends but not during the week. In online supplementary appendix 3, figure A3 shows inequalities in step count using the same methodology shown for sedentary time in figure 1. The results for step count mirror those for sedentary time although with slightly weaker relationships.

## DISCUSSION

Among older, retired adults we find consistent evidence that socially disadvantaged people with respect to nearly all SEP measures are more sedentary than their advantaged counterparts. In contrast, there was little evidence of a consistent relationship between SEP and sedentary time averaged over the week for those still employed.

We identified only five previous studies that measured sedentary behaviour objectively in older adults and included measures of SEP.[24–28] All of these used ActiGraph accelerometers worn on the hip, which are unable to

consistently distinguish sitting from standing.[29] Van Holle et al included measures of education, occupation and income but did not examine their association with sedentary behaviour.[25] The other three studies present a mixed picture. Of the three that included a measure of education, higher educational attainment was significantly associated with lower sedentary time in two,[27 28] while the more educated were more sedentary in the other,[26] although the significance of the latter was not clear. Three studies included occupation, grade or social class,[24 27 28] but there was no significant association with sedentary behaviour in any of them. One study included income that was not significantly associated with sedentary behaviour.[26]

Aside from the different measurement of sedentary behaviour, there are possible explanations for these differences. In the study by Dunlop et al,[26] 47% of the National Health and Nutrition Examination Survey (NHANES) sample were aged 60–69 years so a large proportion would still be employed and, if the pattern observed here is true of the USA, this might obscure the relationship among the retired. In the study by van der Berg et al[27] the occupational classification was relatively crude with little discrimination among the women, 69% of whom were homemakers. Sartini et al[24] used a dichotomised social class as a covariate in mutually adjusted models containing a large number of variables several of which might plausibly lie on the causal pathway.

Investigations of the influence of neighbourhood SEP among older adults is limited to studies of self-reported sedentary behaviour in which neighbourhoods have been dichotomised into low and high income.[17 21] We are not aware of any studies investigating the influence of housing tenure, car ownership or subjective social position on sedentary time.

The most important result of our study is that social inequalities in sedentary behaviour averaged over the week are greater in retired people than among the employed. This may be partly due to how employment constrains people's behaviour. Studies of working age adults, such as Van Dyck et al[30], suggest that white collar workers may in fact be more sedentary. Other research has shown that on retirement manual workers lose activity gained from work which is not compensated for by increasing leisure time activity.[42] Given that our study was designed to investigate sedentary behaviour in older people, we only have a small sample of employed people, and this should be investigated further in a working age cohort.

We find relationships even with measures set relatively early in life such as school leaving age, suggesting that the pathways between SEP and sedentary behaviour may be established early in life.

Evidence on why socioeconomically disadvantaged people are more likely to be sedentary in retirement is sparse, and explanations are likely to be a complex interplay of social, physical, cultural environments and health.[9 43] In addition, while it is relatively well established within the physical activity literature that socioeconomically disadvantaged people are less active, there has been little exploration of the pathways.[44] Qualitative research indicates health is a key determinant of sedentary behaviour,[45] and there is evidence that obesity may cause sedentary behaviour[46] rather than necessarily be a consequence. However, for these cohorts, health and obesity are more likely to mediate the relationship between SEP and sedentary behaviour than be a confounder. The obesogenic environment that had led to socio-inequalities in obesity was not well established until these cohorts were in midlife or later.[47] This is after most life course critical periods of social mobility during which obesity could determine socioeconomic position.

One possible explanation for these results is that workers in more physically active occupations are less likely to develop active leisure activities during working life and this carries on into retirement.[42] Additionally, a qualitative study has indicated that financial costs may prevent people taking part in activities that encourage people to be active.[48]

## Strengths and limitations

Our study has the most comprehensive range of SEP measures in any investigation of sedentary behaviour that we are aware of. The measures are made at the individual, household and area level and pertain to different stages throughout the life course, some quite distal to the outcomes. The area level measures are obtained by geocoding the participant's residential address and so are objective. Even those that were self-reported were ascertained prior to, and independently of, the measurement of sedentary behaviour. Finding consistent results across such a wide range of measures suggests that our results are unlikely to be due to chance.

We used the activPAL3 monitor that provides an objective measure of sedentary behaviour that correctly identifies posture. The activPAL is also worn continuously, whereas hip worn monitors are typically removed at night, and when showering or bathing, which introduces additional sources of error.

The data cover an entire 7-day period, thus minimising any systematic variation over the course of the week. A full week of data were available for a very high proportion of the participants.

We used the SII to facilitate comparison of SEP indicators made on different scales.

Sampling from existing cohorts also has its drawbacks. As longitudinal studies progress, they increasingly suffer from attrition and survival bias. Furthermore, the sample included here is generally more advantaged than the cohorts from which they were drawn. However, it is unclear whether this will have biased the relationships estimated here. In this paper, we have focused on an overall measure of sedentary behaviour. It is possible that information on the context in which sedentary behaviour occurs would provide the detail needed to explain the patterns observed here.[49]

Another weakness of our approach is the reliance on self-reports of sleep and waking times. Efforts to accurately identify sleep time from accelerometry data might prove fruitful for future research.

## CONCLUSIONS

In conclusion, sedentary behaviour appears to be socially patterned among older people after retirement but not before. Prior to retirement, the constraints of the workplace may be masking effects, which are only apparent at weekends. The results here reinforce the message that retirement is a key transition and an opportunity for interventions to improve health and lessen health inequalities. Policies to address health inequalities in later life should provide opportunities and support for older people to develop habits and leisure time activities that replace sedentary behaviour.

**Acknowledgements** The named authors present the study on behalf of the Seniors USP Team, which comprises: Dawn A Skelton (PI), Sebastien Chastin, Simon Cox, Elaine Coulter, Iva Čukić, Philippa Dall, Ian Deary, Geoff Der, Manon Dontje, Claire Fitzsimons, Catherine Gale, Jason Gill, Malcolm Granat, Cindy Gray, Carolyn Greig, Elaine Hindle, Karen Laird, Gillian Mead, Nanette Mutrie, Victoria Palmer, Ratko Radakovic, Naveed Sattar, Richard Shaw, John Starr, Sally Stewart and Sally Wyke. The Lothian Birth Cohort 1936 (LBC1936) would like to thank the cohort members, investigators, research associates and team members. We also thank the radiographers at the Brain Research Imaging Centre and the research nurses and Genetics Core staff at the Wellcome Trust Clinical Research Facility. The Twenty-07 study would like to thank all of the cohort participants and the survey staff and research nurses who carried out the study. The data are employed here with the permission of the Twenty-07 Steering Committee.

**Contributors** SFMC, PMD, IJD, GD, CRG and DAS contributed to the design of the study. GD conceived original idea for the paper and provided statistical support. RJS conducted the analyses and wrote the first draft of the manuscript. IČ assisted with preparation of LBC1936 data. IJD is director of the LBC1936 study. CRG provided statistical advice. IČ, IJD and CRG drafted LBC1936 methodology. PMD and SFMC contributed to acquisition, analysis and processing of activPAL data. DAS was the principal investigator for Seniors USP and provided gerontological advice. All authors have read and commented on the manuscript and approved the final version.

**Funding** The Seniors USP (understanding sedentary patterns) project is funded by the UK Medical Research Council (MRC) as part of the Lifelong Health and Wellbeing Initiative (LLHW) [MR/K025023/1]. The West of Scotland Twenty-07 Study was funded by the MRC and the data were originally collected by the MRC Social and Public Health Sciences Unit (MC_A540_53462). LBC1936 data collection are supported by the Disconnected Mind project (funded by Age UK and MRC [Mr/M01311/1 and G1001245/96077]) and undertaken within the University of Edinburgh Centre for Cognitive Ageing and Cognitive Epidemiology (funded by the BBSRC and MRC as part of the LLHW [MR/K026992/1]).

**Competing interests** IJD is supported by Age UK. PMD has received grant funding from PAL technologies outside the submitted work.

**Patient consent** Obtained from patients.

**Ethics approval** Ethics approval for the Twenty-07 West of Scotland study participants was gained for each wave from the NHS and/or Glasgow University Ethics Committees. Ethical approval for LBC1936 was obtained from the Multi-Centre Research Ethics Committee and from Lothian Research Ethics Committee.

**Provenance and peer review** Not commissioned; externally peer reviewed.

**Data sharing statement** Data collected as part of the Seniors USP study are embargoed until October 2018. Thereafter data sharing will be governed by the agreements already in place for the Twenty-07 and LBC1936 studies. The West of Scotland Twenty-07 study is managed by the MRC/CSO Social and Public Health Sciences Unit, University of Glasgow. Further information about how to access the data can be found at http://2007study.sphsu.mrc.ac.uk/. Data are available upon request from the Lothian Birth Cohort 1936 Study. To request the data, readers should contact the principal investigator, Ian Deary, who can be contacted at i.deary@ed.ac.uk.

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
