## [Reviewer comments · BMJ Open]

ARTICLE DETAILS

TITLE (PROVISIONAL)	Relationships between socioeconomic position and objectively measured sedentary behaviour in older adults in three prospective cohorts
AUTHORS	Shaw, Richard; Ćukić, Iva; Deary, Ian; Gale, Catharine; Chastin, Sebastian; Dall, Philippa; Skelton, Dawn; Der, Geoff

VERSION 1 - REVIEW

REVIEWER	Mark Hamer Loughborough, UK
REVIEW RETURNED	27-Feb-2017

GENERAL COMMENTS	This paper examines social patterning of sedentary behaviour. It is unique in that sedentary behaviour is measured objectively using a postural allocation technique that presumably overcomes limitations of other measures. I have several comments to help improve the paper. 1. Methods: The analytic sample clearly reflects subsets of participants from the overall cohorts involved. It is unclear how representative they are of the overall cohorts. Please clarify. The device is worn continuously over the 7 day period but how did the analyses handle participants who only wore the device for part of the full wear protocol? How was 'waking time' derived. I presume from subtracting the self reported sleep time or was an algorithm employed to derive sleep time from waking time?2. I was surprised that coefficients are not adjusted for potential confounders. In particular morbidity and obesity are likely to explain associations between SEP and sedentary. The authors should clarify/justify why covariates were not included in the models. Were any analyses performed using stepping data instead of sedentary. It would be interesting to see if the same SEP trends are seen for stepping in this data set.3. Have the authors considered examining these data in the context of cross cohort comparisons? For example in Table 2 it seemed that there was a steeper gradient between lifetime social class and sedentary in the 1930s compared with 1950s cohort.4. The paper identifies four previous studies that measured sedentary behaviour objectively in older adults and included measures of SEP. However, data from Whitehall II also examined these associations using actigraph [https://www.ncbi.nlm.nih.gov/pubmed/22791800] and wrist worn Geneactiv device [https://www.ncbi.nlm.nih.gov/pubmed/24500862]. please consider citing them.5. Although having an objective assessment of sedentary is an advantage, it is also a disadvantage not to have context specific data on different types of sedentary behaviours. For example, it is
---

	unsurprising that trends were not picked up in working adults because higher SEP participants will likely have office based jobs and a much smaller proportion of their sitting hours are discretionary, thus vastly reducing individual variability. 6. There are no real explanations offered in the discussion to interpret the results. why do more socially disadvantaged people sit more?? Is it simply being explained by confounding?
--	---

REVIEWER	Olga Theou Dalhousie University, Canada
REVIEW RETURNED	08-Mar-2017

GENERAL COMMENTS	Thank you for the opportunity to review this very interesting study. Please see below some suggestions that could improve the clarity of your manuscript. Abstract “For example, the social class difference in percentage waking time sedentary was -0.5% (95% CI -9.0 to 8.0) for employed people in the Twenty-07 1950s cohort as opposed to 5.7% (95% CI 2.6% to 8.7%) for the retired people in all cohorts combined.” The % difference is not clear. Who had 0.5% less sedentary time the least socially deprived or the most socially deprived? Please clarify. “To be eligible, participants had to complete sleep diaries.” Are these the eligibility criteria for your study or for the cohort studies included? In the methods section please comment whether the sample was representative of the general population of the area or whether it was a convenient sample. Were the data about SEP and sedentary behavior collected similarly among the three cohorts or were there any difference in how questions were asked and response options? Please provide more details about the ActivPAL device. How often data points were collected? How were data processed and analyzed? Provide more information about the missing data (9% of your sample)? How many participants were excluded because of missing ActivPAL data? Did any participant remove the device? Did any device stop working during data collection? Was this 9% of the sample different from the rest of the group regarding demographics and SEP measures? Sedentary time can be different between weekdays and weekends especially for employed participants? As a sensitivity analysis it could be interesting to repeat your analysis separate for weekend and weekdays and examine whether this may have an impact on the findings. Provide more details about how “employed” versus “retired” was defined. How was this determined? Did people have to work full time? What about volunteer positions? Were the questions similar for all cohorts? Table 1: add %females/males
--

	Tables: Can you please indicate which differences are statistically significant? Figure 1: It is not clear the direction of the difference.
--	---

VERSION 1 – AUTHOR RESPONSE

Thank you for the opportunity to review this very interesting study. Please see below some suggestions that could improve the clarity of your manuscript.

Abstract “For example, the social class difference in percentage waking time sedentary was -0.5% (95% CI -9.0 to 8.0) for employed people in the Twenty-07 1950s cohort as opposed to 5.7% (95% CI 2.6% to 8.7%) for the retired people in all cohorts combined.” The % difference is not clear. Who had 0.5% less sedentary time the least socially deprived or the most socially deprived? Please clarify.

“To be eligible, participants had to complete sleep diaries.” Are these the eligibility criteria for your study or for the cohort studies included?

In the methods section please comment whether the sample was representative of the general population of the area or whether it was a convenient sample.

Were the data about SEP and sedentary behavior collected similarly among the three cohorts or were there any difference in how questions were asked and response options?

Please provide more details about the ActivPAL device. How often data points were collected? How were data processed and analyzed?

Provide more information about the missing data (9% of your sample)? How many participants were excluded because of missing ActivPAL data? Did any participant remove the device? Did any device stop working during data collection? Was this 9% of the sample different from the rest of the group regarding demographics and SEP measures?

Sedentary time can be different between weekdays and weekends especially for employed participants? As a sensitivity analysis it could be interesting to repeat your analysis separate for weekend and weekdays and examine whether this may have an impact on the findings.

Provide more details about how “employed” versus “retired” was defined. How was this determined? Did people have to work full time? What about volunteer positions? Were the questions similar for all cohorts?

Table 1: add %females/males

Tables: Can you please indicate which differences are statistically significant?

Figure 1: It is not clear the direction of the difference.

VERSION 2 – REVIEW

REVIEWER	Mark Hamer Loughborough University, UK
REVIEW RETURNED	08-May-2017

GENERAL COMMENTS	The authors have thoroughly addressed all issues
--

REVIEWER	Olga Theou Dalhousie University, Canada
REVIEW RETURNED	11-May-2017

GENERAL COMMENTS	To provide additional information about the feasibility of using accelerometers/inclinometers, please add the following information from your response to the manuscript "two participants were excluded because they did not return activPAL devices. Seven were excluded because there was incomplete sleep diary data. Five were excluded due to poor activPAL quality, assessed using a graphical inspection of the data and 59 due to having sufficient days of data. The reasons for insufficient days of data were varied and not always reported. Eight removed devices due to skin irritation, in other cases the device had fallen off, become wet, or removed for a variety of other reasons. "
--